# Meet in the Middle: A New Pre-training Paradigm

Anh Nguyen*   Nikos Karampatziakis*   Weizhu Chen

Microsoft Azure AI

## Abstract

Most language models (LMs) are trained and applied in an autoregressive left-to-right fashion, predicting the next token from the preceding ones. However, this ignores that the full sequence is available during training. In this paper, we introduce "Meet in the Middle" (MIM) a new pre-training paradigm that improves data efficiency by training in two directions, left-to-right and right-to-left, and encouraging the respective models to agree on their token distribution for each position. While the primary outcome is an improved left-to-right LM, we also obtain secondary benefits in the infilling task. There, we leverage the two pre-trained directions to propose an infilling procedure that builds the completion simultaneously from both sides. We conduct extensive experiments on both programming and natural languages and show that MIM significantly surpasses existing pre-training paradigms, in left-to-right generation as well as infilling. [2]

## 1   Introduction

Language models (LMs) are becoming essential for various assisted authoring tasks, including text summarization and code completion. In order to be usable in many different applications, most LMs must be able to generate the next token from the sequence of previous tokens. Thus, pre-training has focused on optimizing the model's ability to predict the next token given the previous tokens (the prefix), as measured by perplexity. However, this neglects that at pre-training time we also have access to the subsequent tokens (the suffix). While the suffix cannot be used as an input to the model, there are other ways to incorporate it into pre-training which have not received much attention in the literature. In this work we propose a more data-efficient approach, preserving the autoregressive nature of LMs while fully utilizing pre-training data.

Why should we be interested in alternative pre-training procedures? After all, the main artifact produced during pre-training is an autoregressive left-to-right LM and the pre-training objective closely matches both how the LM is applied and the desire to predict the next token well. Nonetheless, there are still compelling reasons to consider alternative training objectives. First, data efficiency: current LMs are trained with sparse supervision, only learning from the actual next token. By exploring denser supervision during training, we aim to improve performance with less data. Second, related tasks: in real-world scenarios, users may need to infill or modify existing text. Current left-to-right LMs are limited in this regard, as they cannot use both sides of the insertion position, often leading to suboptimal results.

In this work, we introduce a novel pre-training paradigm named "Meet in the Middle" (MIM). The MIM approach is founded on two central principles. Primarily, it introduces an additional right-to-left LM to complement the standard left-to-right model, enabling both models to co-regularize and benefit from each other's context. This improves data efficiency and consistency, with models metaphorically meeting in the middle by adjusting their output probabilities to agree with one another. Secondarily,

---

*Equal Contribution

[2]Code and models available at `https://github.com/microsoft/Meet-in-the-Middle`

MIM enables an efficient infilling inference procedure that capitalizes on the agreement between the two LMs developed during the pre-training phase. For infilling, the models literally meet in the middle, constructing the completion from both ends. This aspect of MIM, while secondary to the pre-training objective, offers crucial benefits. Co-regularization not only reinforces consistency between the two LMs but also aids in early termination of the generation process during infilling.

To train MIM, we employ two decoding flows within a single shared decoder-only architecture [BMR$^+$20], [CND$^+$22]. The forward and backward LMs generate tokens in opposite directions, predicting the next token and previous token, respectively. We pre-train both models jointly on a large text corpus using a combination of standard LM loss and an agreement regularizer. Upon completion, the forward model serves as a drop-in replacement for existing autoregressive LMs, while the backward model can be discarded or utilized for tasks such as infilling.

In our experiments, we assess MIM's effectiveness for pre-training LMs across various domains and tasks. Using public code and language data, we pre-train LMs of different sizes and evaluate their performance based on perplexity and code completion tasks. We compare MIM with FIM [BJT$^+$22] and other baselines, demonstrating its superior performance in terms of perplexity and task-specific evaluation metrics. Furthermore, we conduct ablation studies to validate the effectiveness of our primary proposals during training and inference. In summary, our main contributions include:

- Introducing a novel pre-training paradigm for LMs that efficiently utilizes training data by incorporating both prefix and suffix context while retaining the autoregressive nature of LMs. This is achieved by training forward and backward models and promoting their agreement.

- Proposing a simple, efficient infilling inference procedure that leverages context from both sides and the tendency of forward and backward models to agree. Our approach employs parallelism more effectively than existing procedures, resulting in superior quality and latency compared to the state of the art.

- Pre-training LMs of various sizes on public code and language data using MIM, evaluating their performance on both human and programming languages, and demonstrating that MIM outperforms numerous baselines in standard evaluation metrics. Additionally, we make some models and code publicly available.

## 2  Preliminaries

We introduce notation used throughout the paper. For a token sequence $x_1, x_2, \ldots, x_N$, we denote $x_{<i}$ as the prefix $x_1, x_2, \ldots x_{i-1}$ and $x_{>i}$ as the suffix $x_{i+1}, x_{i+2}, \ldots x_N$. Definitions for $x_{\leq i}$ and $x_{\geq i}$ are analogous. We suppress model dependencies on learnable parameters. Arrows distinguish the two models and their outputs; for example, $\overrightarrow{p}$ represents the forward model and $\overleftarrow{p}$ the backward model, while $\overrightarrow{H}$ and $\overleftarrow{H}$ are hidden representations from the respective models.

### 2.1  The Infilling task

Given a token sequence $x_1, x_2, \ldots, x_N$, an insertion position $i$, the infilling task involves generating a plausible $M$-token sequence $y_1, \ldots, y_M$ to fill the gap between $x_{\leq i}$ and $x_{\geq i+1}$. In real-world applications, $M$ is unknown. The infilling task is harder than the left-to-right generation task as the latter is a special case of infilling with an empty suffix. Since generating the best $M$-token left-to-right generation is a search problem that is tackled by heuristics such as greedy search or beam search, we will also follow heuristic approaches for the strictly harder infilling problem.

A technique for infilling that utilizes context from both sides is "Fill in the Middle" (FIM) [BJT$^+$22]. FIM forms the context by concatenating the suffix and prefix (in that order), ensuring coherence near the completion point. FIM is easily applied to pre-trained LMs with minimal modification and is computationally efficient. However, FIM's drawbacks include training on unnatural concatenations of prefix and suffix, giving the prefix more influence on the completion over the suffix, and an ad-hoc training procedure that only considers a few of the $O(N^2)$ possible splits in an $N$-token document.

## 2.2 Bidirectional Language Modeling

Bidirectional language modeling has been used in the literature in two different ways: The first is to describe non-autoregressive LMs with objectives like Masked Language Modeling, which produce better representations but face difficulties in in-context learning [PLR$^+$22].

Earlier work [HR18, PNI$^+$18] used the term bidirectional language modeling to describe models with left-to-right and right-to-left autoregressive flows. Our approach falls in this category, remaining autoregressive and using future tokens only for regularization during training.

# 3 Meet in the Middle

Our proposal involves pre-training two models to predict their next token from different input views (prefix vs. suffix), while maintaining agreement on the probability distributions for these tokens. This provides a dense supervision signal compared to simply predicting the next token. It also encourages the models to "meet in the middle" i.e. reach a compromise prediction informed by different views.

## 3.1 Pre-training

We train two decoder-only LMs, a forward model $\overrightarrow{p}$ and a backward model $\overleftarrow{p}$, with shared parameters. $\overrightarrow{p}$ predicts next tokens in the forward direction, while $\overleftarrow{p}$ predicts previous tokens in the backward direction. To improve data efficiency, we use a co-regularization term that encourages the models to agree on the predicted probability distribution over the vocabulary for each token. We use total variation distance to measure disagreement between the models. Define the disagreement on $x_i$

$$D_{i,x}^{TV}(\overrightarrow{p}||\overleftarrow{p}) = \frac{1}{2}\sum_{z \in \mathcal{V}} |\overrightarrow{p}(z|x_{<i}) - \overleftarrow{p}(z|x_{>i})|$$

where $\mathcal{V}$ is the vocabulary. The agreement regularizer is the sum of these terms over all sentences and tokens. It provides the models with a denser supervision signal which improves data efficiency and helps us train a better autoregressive LM. This is because two probability distributions over all tokens are compared while in the traditional LM objective only the probability of the observed token matters. Moreover it encourages the models to agree on their predictions which helps with the efficiency of infilling as we will see later. Then, the full training loss for dataset $S$ of sequences is given by:

$$\sum_{x \in S} \sum_{i=1}^{|x|} -\log\left(\overrightarrow{p}(x_i|x_{<i})\right) - \log\left(\overleftarrow{p}(x_i|x_{>i})\right) + \beta \cdot D_{i,x}^{TV}(\overrightarrow{p}||\overleftarrow{p}). \tag{1}$$

The hyperparameter $\beta$ is set to $0.1$ in our experiments. After pre-training, $\overrightarrow{p}$ can be used as a left-to-right autoregressive LM.

## 3.2 Infilling

### 3.2.1 Inference

Our inference procedure for infilling aims to be efficient and low-latency. Our approach is shown in Figure 1. At a high level, the two models start building a completion each from their own side and they try to literally meet in the middle. Initially, the prefix and suffix are consumed by $\overrightarrow{p}$ and $\overleftarrow{p}$ respectively. Then $\overrightarrow{p}$ and $\overleftarrow{p}$ generate tokens synchronously one at a time. For each generated token from $\overrightarrow{p}$ (resp. $\overleftarrow{p}$) we check whether it matches a generated token from $\overleftarrow{p}$ (resp. from $\overrightarrow{p}$). If there is a match, we have a "meet in the middle" candidate position for joining the two generated sequences.

In an ideal scenario, $\overrightarrow{p}$ and $\overleftarrow{p}$ generate identical sequences, albeit in reverse order, resulting in "meeting in the middle" after half of the tokens are generated by each model. The agreement regularizer is thus crucial; if $\overrightarrow{p}$ and $\overleftarrow{p}$ yield disparate sequences, our method's speed is inferior to FIM since early termination isn't possible. However, if there's complete agreement, with sufficient parallelism, our method can outperform FIM in generation latency, as each model only needs to generate half the completion autoregressively. Our experiments demonstrate our method's superior latency to FIM, indicating that in most instances, sequences meet close to the middle.

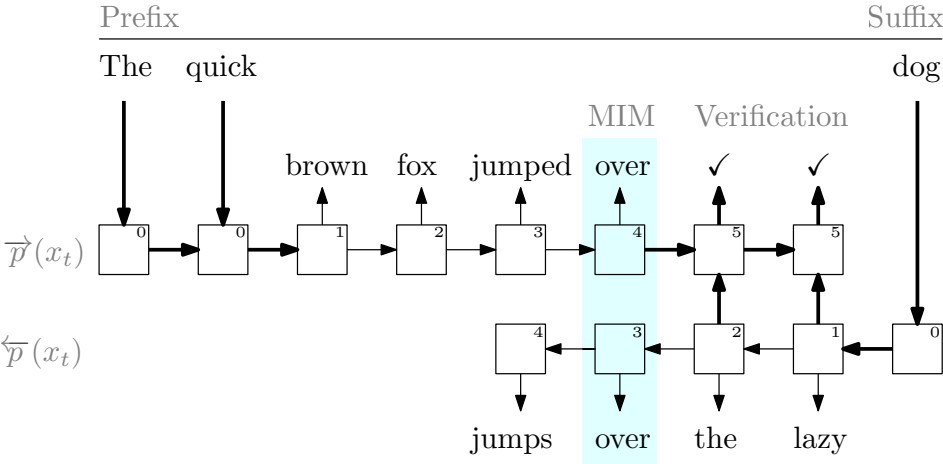

Figure 1: Inference procedure for infilling. Given the prefix "The quick" and the suffix "dog", the models $\overrightarrow{p}$ and $\overleftarrow{p}$ generate tokens until a candidate meet-in-the-middle token is detected (shaded area). We use a single token for illustration purposes, although the method can use more tokens. Given the candidate MIM token, the $\overrightarrow{p}$ (respectively $\overleftarrow{p}$) model can in parallel verify that the tokens generated by $\overleftarrow{p}$ (resp. $\overrightarrow{p}$) are acceptable completions (only the $\overrightarrow{p}$ verification is shown to reduce clutter). The numbers in the top right of each box show the order of operations. Two boxes with the same number can be executed in parallel. Similarly, thick lines show that information (embeddings, tokens) can flow in parallel, while thin lines denote sequential steps.

However, the procedure may yield false positives. For instance, matching a token such as "the" from both sides may not guarantee a coherent infilling upon joining. To mitigate this, we employ 4-gram matching in our infilling experiments. Additionally, we run a parallel verification procedure to ensure that the resultant sequence could have been generated by either $\overrightarrow{p}$ or $\overleftarrow{p}$ running to completion.

Our parallel verification procedure is adapted from [GXS$^{+}$22]. Concretely, assume that the latest token generated from $\overrightarrow{p}$ matches with a token in position $s$ generated from $\overleftarrow{p}$, as is the case in Figure 1 for the token "over". Then we can use the tokens that were generated by $\overleftarrow{p}$ before $s$ as inputs to $\overrightarrow{p}$ *in parallel*. In the context of Figure 1 we provide the tokens "over", "the", "lazy" as inputs to $\overrightarrow{p}$ in parallel. If each output of $\overrightarrow{p}$ matches the corresponding input, we have verified that $\overrightarrow{p}$ would have autoregressively generated the tokens it copied from $\overleftarrow{p}$'s output. Grounding the discussion back to Figure 1, if $\overrightarrow{p}$ generates (in parallel) "the" and "lazy" we have verified that $\overrightarrow{p}$ would have autoregressively generated the same tokens. In cases of partial or no match, the generation from $\overrightarrow{p}$ either fast-forwards to the first disagreement point or resumes autoregressive generation from $\overrightarrow{p}$ and $\overleftarrow{p}$. While our implementation use greedy top-1 sampling and comparison of input and output at each position, we could adopt less stringent acceptance criteria, like a sufficiently high probability of the candidate token in the previous output. If $\overrightarrow{p}$ and $\overleftarrow{p}$ finish generating without meeting or passing verification, we return the higher-probability sequence according to its generating model.

### 3.2.2 Infilling Specific Enhancements

Our model's infilling performance can be boosted by adopting the Synchronous Bidirectional Attention mechanism [ZZZ19]. Though this reduces compatibility with autoregressive language models (LMs), it enhances generation through bidirectional conditioning. The Synchronous Bidirectional Attention layer modifies the regular attention layer activations as follows: Let $\overrightarrow{H}$ and $\overleftarrow{H}$ be the outputs of an attention layer in $\overrightarrow{p}$ and $\overleftarrow{p}$ respectively. For a hyperparameter $\lambda$, the fused attention hidden representation becomes $\overrightarrow{H}_f = \overrightarrow{H} + \lambda\overleftarrow{H}$ and $\overleftarrow{H}_f = \overleftarrow{H} + \lambda\overrightarrow{H}$. Instead of $\overrightarrow{H}$ and $\overleftarrow{H}$, the models $\overrightarrow{p}$ and $\overleftarrow{p}$ use $\overrightarrow{H}_f$ and $\overleftarrow{H}_f$. If $\lambda = 0$, $\overrightarrow{p}$ and $\overleftarrow{p}$ become classic autoregressive transformers. However, careful training is necessary to avoid information leakage with this layer. To mitigate this, we adopt a two-stage training process, see [ZZZ19] for details.

# 4 Experiments

## 4.1 Data and Models

Our code models are pre-trained on a large and diverse corpus of public code with permissive licenses, covering multiple programming languages. The dominant languages in our corpus are Python, Java, and C++. After filtering and deduplication, the corpus contains about 300B tokens. Details are provided in Table 9. Our models train on a single pass over this data. This is about six times larger than the dataset used to train the Incoder model [FAL+22], which included 50B tokens comprising code and Stack Overflow data.

We train our natural language models on data from the following datasets: **CC-News**, **OpenWebText**, **CC-Stories**, and **CC-100** to assess the models language modeling capability. In particular:

- **CC-News**: 63M English news articles crawled from Sept 2016 to Feb 2019 (76GB).

- **OpenWebText**: An open source recreation of the GPT-2 WebText dataset (38GB).

- **CC-Stories**: A subset of CommonCrawl data filtered to match the story-like style of Winograd schemas (31GB).

- **CC-100**: A dataset extracted from CommonCrawl snapshots between Jan 2018 and Dec 2018, filtered to match the style of Wikipedia (292GB).

Apart from the natural language data utilized in our language modeling experiments, we also train our models on a filtered version of the Falcon RefinedWeb corpus [PMH+23] used in the previous work of [LBE+23], that has a total of 88B tokens. We then present the performance results on common natural language benchmarks in section 4.3.3

We use a decoder-only transformer LM [VSP+17, BMR+20] for pre-training. The model is trained to predict the next token in both left-to-right and right-to-left directions, using the same model and parameters. To explore the impact of model size, we pre-train models with different capacities: 350M, 1.3B, 2.7B. For the hyperparameters and training setup of each model size, please refer to the Appendix.

As baselines, we use well-known published results but also pre-train three FIM models with context-level FIM [BJT+22]. Character-level transformations are applied, with a FIM-rate of $0.5$ and SPM+PSM joint training. Both the MIM and FIM models are pre-trained using the Megatron-LM framework [SPP+19]. See the Appendix for model configurations and architectural details.

## 4.2 Benchmarks and metrics

### 4.2.1 Code generation and infilling

We assess MIM's performance both in autoregressive left-to-right generation and infilling tasks. For the autoregressive generation, the model generates the code body from a function signature, docstring, and test cases. Evaluation datasets include **HumanEval** [CTJ+21], **MBPP** [AON+21], and **APPS** [HBK+21] for Python problems. We use the multilingual **HumanEval-X** dataset for multiple programming languages. For this task, the left-to-right model alone is used for inference without any provided suffix.

In the infilling task, the model fills blank lines in an incomplete program. This is evaluated using the **HumanEval Infilling** [BJT+22] and **MBXP** [AGW+22] datasets. MIM is compared with Incoder, three FIM variants we trained on the same data as MIM, and code-davinci-002, as these datasets provide suffixes that not all models can manage effectively.

We use the pass@k metrics [CTJ+21] to measure the percentage of generated codes that pass all tests within the top-k candidates. We report pass@1, pass@10, and pass@100. For generation, top-p sampling with $p = 0.95$ is employed with temperatures of $0.2$ for pass@1 and $0.8$ for pass@10 and pass@100. We also include single-line exact match (EM) metrics to indicate the percentage of completed lines matching the reference solution's masked lines [FAL+22, BJT+22].

### 4.2.2 Language Modeling

In addition to code generation and infilling, we evaluate MIM's language modeling capabilities, particularly its next-token prediction ability measured by perplexity. Perplexity is evaluated in both **in-domain** and **out-of-domain** settings. The former involves a held-out subset of the combined training data, while the latter uses the Pile dataset [GBB+21], a multi-domain public language modeling dataset. Average perplexity across all Pile subsets is reported for our models and baselines.

### 4.2.3 Natural Language Benchmarks

Finally, we evaluate MIM on established benchmarks for natural language, encompassing both commonsense reasoning and language understanding. For commonsense reasoning, we have tested our models on Winogrande, ARC-Easy, ARC-Challenge, BoolQ, and SIQA. In terms of knowledge and language understanding, we have assessed MIM's performance on PIQA, HellaSwag, and OpenbookQA. We compare against **phi-1.5-web-only** [LBE+23] and present zero-shot results for these tasks using the LM-Eval Harness, with the exception of MMLU where we use 2-shot performance.

## 4.3 Main results

### 4.3.1 Code generation and infilling

First we focus on the superior performance of our from-scratch pre-trained FIM model over various baselines, in Table 1. For instance, our 2.7B model yields a 28.5% pass@1 in HumanEval, significantly outperforming the 15.2% pass@1 of Incoder's 6.7B model [BJT+22]. FIM-2.7B also beats other strong baselines such as Codex 2.5B and CodeGen-Multi-6.1B, achieving 67.8% in pass@100 compared to their respective 44.9% and 59.5%. This performance is attributable to larger, cleaner training data and implementation details (discussed in the Appendix).

Having established a strong baseline, we now benchmark MIM's autoregressive generation in a prefix-only setting using both the HumanEval and MBPP datasets. While FIM vs. MIM is the apples-to-apples comparison, we include many other models for reference and note that all these could have leveraged our MIM objective during their training. Table 1 shows MIM consistently outperforming FIM across all metrics for all model sizes (350M, 1.3B, 2.7B). For example, MIM-2.7B enhances HumanEval pass@1 to 30.7%, a 2.2% gain over FIM-2.7B. The improvement on MBPP is similar, in terms of pass@1. We note that the improvements increase with the model size. This is evidence that the larger models benefit more from the agreement regularizer. We further evaluate this setting on the APPS dataset, which has three different difficulty levels. We report the results in Table 3. The trend and improvement are similar to the other two datasets, in which MIM consistently outperforms FIM across all the metrics and difficulty levels. In the multilingual setting, we compare MIM with FIM baselines in the HumanEval-X dataset. We observe consistent improvement across all metrics and all the programming languages that we evaluated on. Results are reported in Table 2.

These findings illustrate MIM's superiority over FIM for left-to-right autoregressive generation. The main claim in [BJT+22] is that FIM does not harm the original left-to-right generative capability and can be learned for free. We contend that MIM's pre-training, enhanced by the agreement regularizer, results in a superior generative model, thereby proposing a new pre-training paradigm.

Finally, we test MIM in an infilling setting, as motivated by real-world applications like GitHub Copilot, where developers frequently edit in the middle of source files. MIM outperforms FIM across all metrics and model sizes on both the HumanEval Infilling and the MBXP Infilling benchmarks (Table 1). Notably, the MIM-2.7B model achieves pass@1 of 26.3% and an exact match metric of 57.8%, improvements of 3.5% and 6.1%, respectively, over the FIM-2.7B model. This underscores the consistent improvement that MIM offers across different model sizes and tasks.

### 4.3.2 Language Modeling

Here we compare FIM and MIM models pre-trained on language modeling tasks. We use perplexity as our evaluation metric and look at two different settings, namely **in-domain** and **out-of-domain**.

The perplexity results of both settings are summarized in Table 4. Across all experiments, MIM consistently outperform FIM baselines in terms of perplexity. The MIM-2.7B model has the best

| Methods | HumanEval | | | MBPP | | | HE Infilling | | MBXP |
|---|---|---|---|---|---|---|---|---|---|
| $k$ | 1 | 10 | 100 | 1 | 10 | 100 | 1 | EM | 1 |
| Incoder-1.3B | 8.9 | 16.7 | 25.6 | 11.3 | 26.8 | 42.7 | 8.6 | 31 | 9.2 |
| Incoder-6.7B | 15.2 | 27.8 | 47 | 19.4 | 46.5 | 66.2 | 14.5 | 44.1 | 20.8 |
| CodeGen-Multi-6B | 18.2 | 28.7 | 44.9 | - | - | - | - | - | - |
| CodeGen-Multi-16B | 18.32 | 32.07 | 50.80 | - | - | - | - | - | - |
| Codex-2.5B | 21.36 | 35.42 | 59.5 | - | - | - | - | - | - |
| Codex-12B | 28.81 | 46.81 | 72.31 | - | - | - | - | - | - |
| code-davinci-001 | 39 | 60.6 | 84.1 | 51.8 | 72.8 | 84.1 | - | - | - |
| code-davinci-002 | 47 | 74.9 | 94.1 | 58.1 | 76.7 | 84.5 | 51.3 | 74 | 57.6 |
| PaLM-8B | 3.6 | - | 18.7 | 5.0 | - | - | - | - | - |
| PaLM-62B | 15.9 | - | 46.3 | 21.4 | - | - | - | - | - |
| PaLM-540B | 26.2 | - | 76.2 | 36.8 | - | - | - | - | - |
| LLaMA-7B | 10.5 | - | 36.5 | 17.7 | - | - | - | - | - |
| LLaMA-13B | 15.8 | - | 52.5 | 22.0 | - | - | - | - | - |
| LLaMA-33B | 21.7 | - | 70.7 | 30.2 | - | - | - | - | - |
| LLaMA-65B | 23.7 | - | 79.3 | 37.7 | - | - | - | - | - |
| FIM-350M | 12.8 | 16.7 | 27.8 | 14.8 | 30.2 | 44.5 | 11.8 | 37.6 | 14.3 |
| FIM-1.3B | 20.8 | 39.4 | 51.7 | 25.9 | 45.6 | 62.5 | 15.7 | 42.2 | 22.1 |
| FIM-2.7B | 28.5 | 45.6 | 67.8 | 38.2 | 61.2 | 76.1 | 22.8 | 51.7 | 30.4 |
| MIM-350M | 13.7 | 17.2 | 28.5 | 16.5 | 33.7 | 47.4 | 14.6 | 41.7 | 16.4 |
| MIM-1.3B | 22.4 | 41.7 | 53.8 | 26.8 | 47.6 | 65.1 | 17.4 | 47.6 | 24.5 |
| MIM-2.7B | **30.7** | **48.2** | **69.6** | **42.2** | **64.8** | **79.3** | **26.3** | **57.8** | **35.7** |

Table 1: pass@k (%) on the HumanEval, MBPP and HumanEval Infilling and MBXP benchmarks. Additionally, Exact Match (EM) metric [FAL$^+$22] is reported for HumanEval Infilling. FIM is the baseline 'Fill in the Middle" (FIM). MIM is our proposed method with the enhancements from section 3.2.2. For reference, we report evaluation numbers of other models, namely Incoder-1.3B and Incoder-6.7B [FAL$^+$22], Codex-12B [CTJ$^+$21], code-davinci-001 and 002. Note that, Codex-12B, davinci-001, PaLM [CND$^+$22] and LLaMA [TLI$^+$23] were trained only with left-to-right autoregressive objective, thus, cannot perform infilling.

| Methods | C++ | | | Java | | | Go | | |
|---|---|---|---|---|---|---|---|---|---|
| $k$ | 1 | 10 | 100 | 1 | 10 | 100 | 1 | 10 | 100 |
| Incoder-6B | 10.0 | 20.0 | 35.0 | 9.0 | 19.0 | 40.0 | 8.0 | 14.0 | 29.0 |
| CodeGen-6B | 12.0 | 20.0 | 36.0 | 15.0 | 18.0 | 40.0 | 9.0 | 22.0 | 40.0 |
| CodeGen-16B | 18.0 | 30.0 | 50.0 | 15.0 | 38.0 | 60 | 13.0 | 25.0 | 47.0 |
| CodeGeeX-13B | 20.0 | 31.0 | 50.0 | 16.0 | 38.0 | 58.0 | 15.0 | 25.0 | 49.0 |
| FIM-350M | 8.5 | 18.3 | 24.3 | 11.3 | 21.5 | 27.6 | 10.2 | 16.8 | 31.4 |
| FIM-1.3B | 16.7 | 31.5 | 43.2 | 15.4 | 25.2 | 32.6 | 12.5 | 23.7 | 39.6 |
| FIM-2.7B | 24.5 | 38.6 | 51.3 | 18.3 | 28.6 | 38.7 | 15.2 | 30.4 | 50.2 |
| MIM-350M | 10.2 | 19.6 | 26.3 | 11.1 | 22.4 | 28.2 | 10.8 | 17.5 | 31.8 |
| MIM-1.3B | 19.3 | 36.5 | 45.7 | 17.6 | 27.4 | 34.7 | 13.6 | 25.1 | 41.7 |
| MIM-2.7B | **27.4** | **41.3** | **54.1** | **21.6** | **30.8** | **39.1** | **17.4** | **32.7** | **53.6** |

Table 2: $pass@k$ (%) results on HumanEval-X in the zero-shot settings for the baselines FIM and our MIM approach. Results of $k = 1$, 10 and 100 are reported across all categories.

perplexity across all datasets in both settings. For example, MIM-2.7B model obtains a perplexity of 9.54 in OpenWebText dataset, a relative 19.9% reduction in perplexity over FIM-2.7B model, which obtains a perplexity of 11.92. In the Pile dataset [GBB$^+$21], which represents the **out-of-domain** setting, MIM-2.7B also outperforms FIM-2.7B by a relative reduction of 15.3% in perplexity (9.24 vs 10.92), which further reinforces the advantages of MIM pre-training for natural languages.

### 4.3.3 Natural Language Benchmarks

We trained a MIM model with a total of 1.3B parameters, which shares architectural details with [LBE$^+$23], on a filtered version of the Falcon RefinedWeb corpus, comprising a total of 88B tokens. This model was subsequently compared with the baseline model, **phi-1.5-web-only** in [LBE$^+$23],

| Methods | Introductory | | | Interview | | | Competition | | |
|---------|:---:|:---:|:---:|:---:|:---:|:---:|:---:|:---:|:---:|
| $k$ | 1 | 10 | 100 | 1 | 10 | 100 | 1 | 10 | 100 |
| FIM-350M | 3.6 | 7.8 | 11.5 | 0.0 | 0.3 | 1.2 | 0.0 | 0.04 | 0.9 |
| FIM-1.3B | 8.2 | 11.6 | 17.4 | 0.12 | 0.59 | 1.9 | 0.01 | 0.07 | 1.7 |
| FIM-2.7B | 12.4 | 15.7 | 20.8 | 0.27 | 0.72 | 2.4 | 0.03 | 0.095 | 2.4 |
| MIM-350M | 4.7 | 9.2 | 14.3 | 0.2 | 0.51 | 2.3 | 0.02 | 0.06 | 1.4 |
| MIM-1.3B | 10.6 | 14.2 | 21.2 | 0.36 | 0.76 | 3.6 | 0.043 | 0.09 | 2.2 |
| MIM-2.7B | **14.3** | **18.2** | **24.6** | **0.52** | **1.4** | **5.2** | **0.067** | **0.18** | **3.3** |

Table 3: $pass@k$ (%) results on the APPS benchmarks in the zero-shot settings for the baselines FIM and our MIM approach. Results of $k = 1$, 10 and 100 are reported across all categories.

| Methods | Datasets | | | | |
|---------|:---:|:---:|:---:|:---:|:---:|
| MODELS | CC-NEWS | OPENWEBTEXT | CC-STORIES | CC-100 | THE PILE |
| FIM-350M | 21.16 | 17.91 | 20.89 | 14.23 | 15.14 |
| FIM-1.3B | 18.78 | 13.28 | 17.52 | 11.34 | 12.48 |
| FIM-2.7B | 13.45 | 11.92 | 13.43 | 9.43 | 10.92 |
| MIM-350M | 19.43 | 17.23 | 18.75 | 13.45 | 13.97 |
| MIM-1.3B | 16.04 | 12.23 | 14.63 | 11.16 | 10.79 |
| MIM-2.7B | **11.17** | **9.54** | **11.35** | **8.76** | **9.24** |

Table 4: Perplexity on all datasets in **in-domain** setting (**CC-News**, **OpenWebText**, **CC-Stories**, & **CC-100** held-out datasets), and perplexity results in **out-of-domain** setting (The Pile [GBB+21])

which was trained on identical data using an autoregressive left-to-right objective. Tables 5 and 6 summarize the results. On commonsense reasoning **MIM-1.3B-web-only** outperforms by 0.42% on average while on knowledge understanding it outperforms by 1.2% on average. Overall, (combining the two tables) MIM improves by 0.76% on average.

## 4.4 Ablation Study

### 4.4.1 Effect of Infilling Specific Enhancements

In this section, we perform an ablation study to qualitatively assess the effect of the enhancements we proposed in section 3.2.2 for the infilling task. We compare the purely autoregressive MIM model ($\lambda = 0$) where the LMs are not allowed to observe generated tokens from the opposite side, and only utilize context from their own side during generation. We contrast this with using the Synchronous Bidirectional Attention layer with $\lambda = 0.3$ that conditions on previously generated tokens from both sides. We used perplexity on the validation data to select the value $\lambda$. We reuse the same value of $\lambda$ during infilling to avoid any potential mismatch between training and inference.

We conduct an experiment on HumanEval Infilling [BJT+22] and results are summarized in Table 7. We notice that models that directly incorporate bidirectional context always outperform models that only utilize unidirectional context across all model sizes. As always, this is at the expense of the forward model no longer being a a drop-in replacement for standard autoregressive LMs.

### 4.4.2 Effect of Agreement Regularizer

While MIM performs better than FIM it's still possible that this difference is simply due to the parameters being trained on two tasks (left-to-right and right-to-left). To preclude this, we perform an ablation study where we remove the token-level agreement regularizer of Section 3.1 during training while keeping the infilling enhancements ($\lambda = 0.3$). We show that encouraging agreement during training helps improve the infilling performance of our models in the HumanEval Infilling benchmark [BJT+22], as summarized in Table 8. Comparing with Table 7, we see that models trained without token-level agreement regularization in general perform worse than models that do not utilize bidirectional context ($\lambda = 0$), which further emphasizes the importance of our agreement regularizer in making the predictions consistent between forward and backward directions.

| Methods | Datasets | | | | |
|---|---|---|---|---|---|
| MODELS | WINOGRANDE | ARC-EASY | ARC-CHALLENGE | BOOLQ | SIQA |
| phi-1.5-web-only | **0.604** | 0.666 | 0.329 | **0.632** | 0.414 |
| MIM-1.3B-web-only | 0.591 | **0.687** | **0.337** | 0.628 | **0.423** |

Table 5: Results of MIM in standard **commonsense reasoning** benchmarks, including **WinoGrande**, **Arc-Easy**, **Arc-Challenge**, **BoolQ** and **SIQA**, vs. the baseline **phi-1.5-web-only** in [LBE$^+$23].

| Methods | Datasets | | | |
|---|---|---|---|---|
| MODELS | PIQA | HELLASWAG | MMLU | OPENBOOKQA |
| phi-1.5-web-only | **0.743** | 0.478 | 0.309 | 0.274 |
| MIM-1.3B-web-only | 0.736 | **0.491** | **0.327** | **0.298** |

Table 6: Results of MIM in standard **knowledge understanding** benchmarks, including **PIQA**, **HellaSwag**, **MMLU** and **OpenbookQA**, vs. the baseline **phi-1.5-web-only** in [LBE$^+$23].

| Methods | | HE Infilling | | MBXP |
|---|---|---|---|---|
| MODELS | $\lambda$ | pass@1 | EM | pass@1 |
| MIM-350M | 0.0 | 12.5 | 38.6 | 14.1 |
| | 0.3 | **14.6** | **41.7** | **16.4** |
| MIM-1.3B | 0.0 | 15.6 | 45.2 | 21.7 |
| | 0.3 | **17.4** | **47.6** | **24.5** |
| MIM-2.7B | 0.0 | 24.7 | 54.3 | 32.4 |
| | 0.3 | **26.3** | **57.8** | **35.7** |

| Methods | | HE Infilling | | MBXP |
|---|---|---|---|---|
| MODELS | Regularizer | pass@1 | EM | pass@1 |
| MIM-350M | no reg | 11.8 | 35.9 | 13.5 |
| | + reg | **14.6** | **41.7** | **16.4** |
| MIM-1.3B | no reg | 13.8 | 42.7 | 22.7 |
| | + reg | **17.4** | **47.6** | **24.5** |
| MIM-2.7B | no reg | 23.2 | 52.5 | 30.6 |
| | + reg | **26.3** | **57.8** | **35.7** |

Table 7: pass@1 (%) results of MIM with ($\lambda = 0.3$) and without ($\lambda = 0.0$) bidirectional context on HumanEval Infilling and MBXP. Exact match results on HumanEval Infilling are also reported.

Table 8: pass@1 (%) results of MIM with and without agreement regularizer on HumanEval Infilling and MBXP. Exact match results on HumanEval Infilling are also reported.

## 4.5 Efficiency

### 4.5.1 Training Efficiency

At first glance, one might hypothesize that our proposed methodology could potentially necessitate twice the computational resources in comparison to the conventional left-to-right autoregressive training, with most of the performance gains attributed to this additional computation. Contrary to this supposition, empirical evidence gathered from our experiments suggests a different narrative. Our Meet-in-the-Middle (MIM) strategy exhibits performance metrics that are not only commensurate but often superior to those of the left-to-right autoregressive training. This assertion is corroborated by a comparative analysis of the validation log perplexity of MIM models and left-to-right autoregressive models across different model sizes. All models were trained on the filtered Falcon RefinedWeb corpus [LBE$^+$23]. As illustrated in Figures 3 and 4 in the Appendix, when assessed under the same computational budget (either FLOP count or wall clock time), MIM models consistently outperform left-to-right autoregressive models across all relevant computational budgets. Besides, when comparing MIM models and left-to-right autoregressive models on a time-basis, we still observe major speedups over the baselines.

### 4.5.2 Inference Efficiency

Finally we assess the efficiency of our infilling procedure, contrasting it with FIM through inference latency across various batch sizes. Figure 2 shows that MIM achieves a speedup over FIM baselines in both single and half precision, using a 1.3B model, across all HumanEval Infilling examples. MIM-1.3B specifically performs 4% to 6% faster than FIM-1.3B in single precision, and 3% to 5% faster in half precision. This is due to MIM leveraging available parallelism better than FIM.

# 5 Related work

There is an extensive body of work on bidirectional language modeling. Early work such as ULMFiT [HR18], and ELMo [PNI+18] kept the autoregressive nature of the LM, while later work such as BERT [DCLT19], T5 [RSR+20], and SpanBERT [JCL+20] focused on representation learning and was non-autoregressive. XLNET [YDY+19], on the other hand, utilizes bidirectional context during training by the permutation language modeling objective, which maximizes the likelihood over all factorization orderings of the training sequences. However, because these later models typically focus on representation learning, in-context learning via prompting can be difficult [PLR+22].

Two works that train neural models using similar ideas are [SKS+18] and [ZWL+19]. In the former the authors train RNNs for real-time acoustic modeling. This constraint is very similar to our desire of having a forward model that can generate the next token from the previous ones. However, they propose to regularize the forward and backward RNNs by requiring their representations to be close in Euclidean distance. We suspect that this constraint may be unnecessarily stringent, and trading it off against the LM's perplexity during training could hurt overall performance.

On the other hand, [ZWL+19] encourage agreement in probability space. However, they are interested only in neural machine translation and only encourage agreement of probabilities at the output sequence level. In contrast, we encourage agreement on every token.

Sharing all parameters between different factorizations of the sequence was first proposed in XLNET [YDY+19] but has also been used with only forward and backward models in the context of image captioning [ZHL+22]. These works motivated us to share all parameters between our two LMs.

Using LMs for infilling was first proposed in [DLL20] where the authors tackled the more challenging setup with multiple blanks. [BJT+22] applies "Fill in the Middle" (FIM) to the training data by randomly splitting each training instance into a tuple of (prefix, middle, suffix) and concatenate these sections into a single example together with their sentinel tokens after tokenization. [BJT+22] also contains a thorough section on research related to infilling. We refer the reader there for more details.

Furthermore, [AHR+22] and [FAL+22] propose an extension of FIM, namely "Causal Masked Language Modeling" (CM3) and explore multi-region infilling problem. Similar to us, [BJT+22] and [FAL+22] leverage FIM to pre-train decoder-only LMs on code data and evaluate their models on zero-shot code completion benchmark created from HumanEval dataset [CTJ+21].

In [SLB17] the authors propose bidirectional beam search that also employs forward and backward LMs. However their focus is in improving accuracy at the expense of latency. Their procedure performs multiple passes over the completion, fixing tokens one-by-one. Our approach focuses on achieving better infilling accuracy than FIM while also reducing latency.

# 6 Conclusion

This paper presented "Meet-in-the-Middle" a novel method devised to address two significant challenges in large LMs: pre-training data efficiency and effective context handling in infilling tasks. By using both forward and backward LMs that share parameters and are trained for mutual agreement and next token prediction, the method significantly enhances the performance of existing autoregressive LMs.

## Acknowledgments

We would like to thank Dejian Yang, Jian-Guang Lou, Yuanzhi Li at Microsoft Research for helping us with training data preparation. We also sincerely thank Daniel Fried and Armen Aghajanyan at Meta AI for helpful discussion regarding the training details of FIM baselines.

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
