# A    Appendix

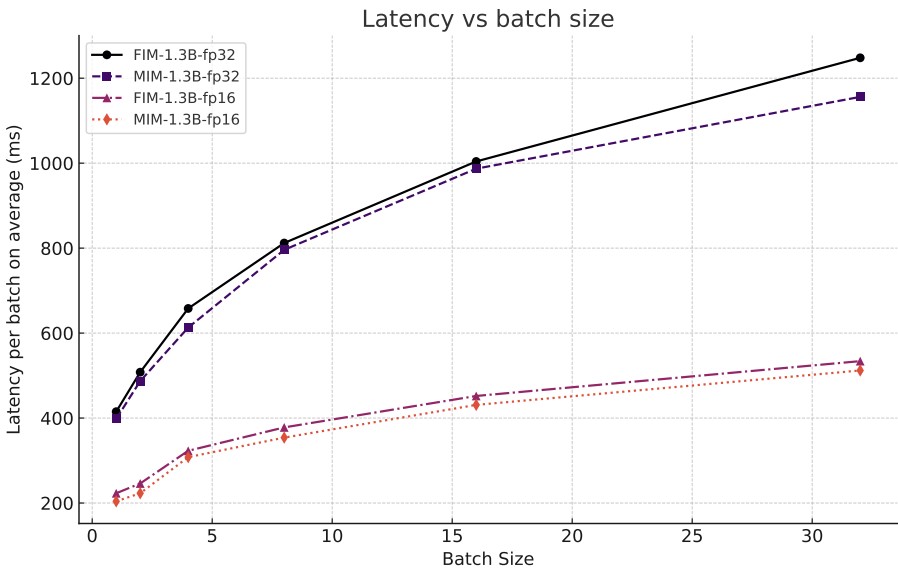

Figure 2: Inference latency with batch implementation in A100 GPU with fp32 and fp16 format.

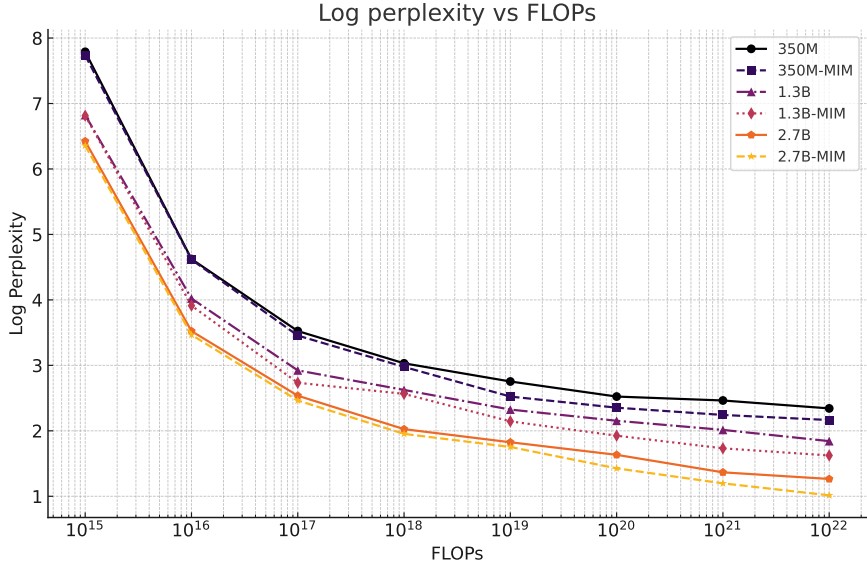

Figure 3: Perplexity vs. FLOP count of MIM compared to left-to-right baselines across model sizes.

## A.1    Model training details

To evaluate the effectiveness of "Meet in the Middle" (MIM) pre-training compared to left-to-right autoregressive and "Fill in the Middle" (FIM) pre-training baselines, we adopt standard transformer-based autoregressive language models used in previous works [BMR+20] for all the models we trained, varying the number of parameters (350M, 1.3B, 2.7B). Moreover, we replace the use of the Multi Head Attention [VSP+17] with the use of the Multi Query Attention proposed in [Sha19] in

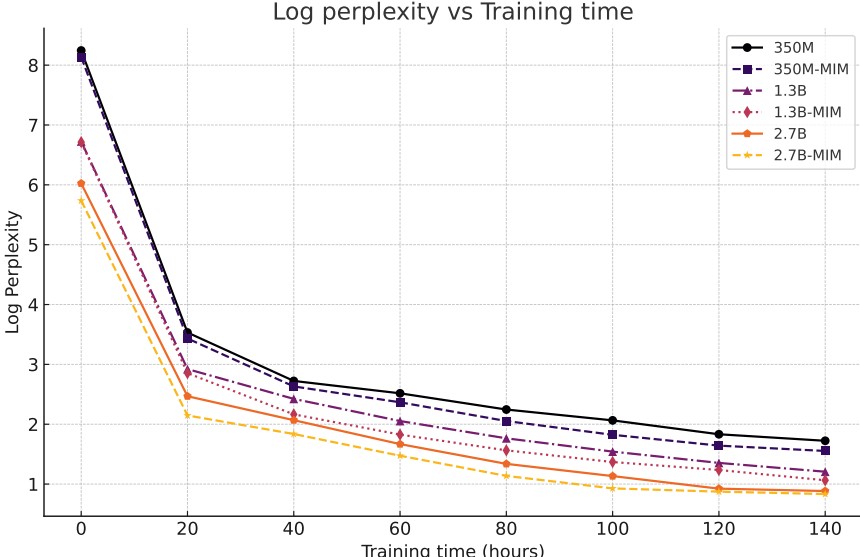

Figure 4: Perplexity vs. training time of MIM compared to left-to-right baselines across model sizes.

all the models we trained, allowing faster inference and reducing the memory requirements to store multiple key and values embeddings that are not shared between attention heads.

For our bidirectional language models, we run the forward model and the backward model in parallel within a single decoder-only architecture, leveraging bidirectional context explicitly during pre-training. We use the sentinel token $\langle l2r \rangle$ to specify that the generation comes from the forward model and sentinel token $\langle r2l \rangle$ to specify that generation comes from the backward model.

Regarding optimization, we use the Adam optimizer [KB15] with $\beta_1 = 0.9$, $\beta_2 = 0.95$, $\epsilon = 10^{-8}$ and a global gradient norm clipping of $1.0$. We follow [BMR+20] to decay learning rate to $10\%$ of its maximum value using cosine annealing with linear warm-up of $2\%$ of the total number of training steps.

For scaling the training of these models, we employ the open source Megatron-LM framework [SPP+19] and partition the training across multiple GPUs along the batch dimension. All the training runs that we conducted use mixed precision training [MNA+18] and FlashAttention [DFE+22] to reduce memory requirements and increase training throughput. During pre-training of our models, we observed that MIM, FIM and autoregressive left-to-right pre-training have similar training wall-clock time, it is because the forward model and the backward model are executed in parallel in MIM pre-training. Our largest models of size 2.7B parameters are trained using 128 A100 GPU with 80GB memory each over 4 days, while the smaller models are trained using 64 A100 GPU with 80GB memory each over 3.5 days. See Table 10 for the details of all the training runs.

## A.2 Programming language dataset details

Table 9 details the statistics of the datasets of different programming languages we use to pre-train our code language models in terms of number of tokens and dataset size. We perform some filtering and deduplication to obtain the final dataset. Our tokenizer is based on the Byte-Pair Encoding algorithm widely used in previous work [CTJ+21] to directly encode raw bytes with a vocabulary of size 100257 tokens. We pre-tokenize the text using a special regex pattern that accounts for splitting on digit and newlines together with the default GPT-2 pre-tokenization [BMR+20].

## A.3 Broader Impact

This paper presents "Meet in the Middle", a novel pretraining paradigm for language models that brings potentially far-reaching benefits to various domains. Enhanced efficiency in training language

| Languages | Size (GB) | Tokens (B) |
|---|---|---|
| C | 34.3 | 12.3 |
| C++ | 215.6 | 70.8 |
| Python | 252.3 | 75.5 |
| Java | 178.5 | 46.7 |
| JavaScript | 120.1 | 39.3 |
| TypeScript | 21.8 | 8.6 |
| PHP | 30.7 | 11 |
| Ruby | 26.8 | 10.1 |
| C# | 35.3 | 12.6 |
| Others | 40.2 | 13.3 |
| Total | 955.6 | 300 |

Table 9: Approximate statistics of the programming language pre-training data

| Hyper-parameters | 350M | 1.3B | 2.7B |
|---|---|---|---|
| Number of layers | 24 | 24 | 32 |
| Number of heads | 16 | 16 | 32 |
| Dimension per head | 64 | 128 | 80 |
| Context length | 2048 | 2048 | 2048 |
| Batch size | 786k | 1M | 1M |
| Weight decay | 0.1 | 0.1 | 0.1 |
| Learning rate | $3e-4$ | $2e-4$ | $2e-4$ |
| Warmup steps | 7k | 5k | 5k |
| Total steps | 382k | 286k | 286k |

Table 10: Details of each training run for all of our model specifications.

models could lead to significant advances in the fields of NLP, machine learning, and AI more broadly. The implications range from cost savings due to more efficient training of these models, reduced environmental impact because of less computational resources needed, to advancements in diverse applications of language models such as translation, chatbots, and text generation.

The proposed method's secondary benefits in the infilling task could also improve several NLP tasks, such as text summarization and question answering, leading to better usability and overall performance of AI systems in these areas. This could bring significant value to industries reliant on these systems, such as customer service, education, and entertainment.

However, as with any improvement in AI language capabilities, there are potential risks and negative implications. For instance, more powerful language models can also be used for manipulative purposes such as generating misleading information or deepfake text content, which could exacerbate the spread of misinformation. Furthermore, as the technology advances, there could be increased societal pressure to use AI in areas where it may not be the best choice due to other considerations, such as privacy or job displacement. Therefore, it is crucial to ensure that the development and application of these advanced models are guided by strong ethical principles and appropriate regulatory oversight.