# OpenReview forum: "Meet in the Middle: A New Pre-training Paradigm"
_NeurIPS.cc/2023/Conference — NeurIPS 2023 poster_

### Official Review · Reviewer_V8eN · 2023-06-11

**Soundness:** 3 good
**Presentation:** 3 good
**Contribution:** 4 excellent
**Rating:** 7
**Confidence:** 4

**Summary:**

The paper proposes a method named "Meet in the Middle" (MIM), an novel inference approach for language models. The uniqueness of MIM lies in its bidirectional training—left-to-right and right-to-left—that not only augments data efficiency but also aligns the token distribution for each position. The method improves the performance on infilling task by employing a dual-direction infilling procedure. Empirical evidence gathered from extensive testing on programming and natural languages substantiates MIM's superior performance over traditional pre-training methodologies in both, left-to-right generation and infilling.

**Strengths:**

1. The novelty of the "Meet in the Middle" (MIM) stands out, particularly with respect to the inference phase in text infilling.

2. MIM successfully addresses the important task of general text infilling, encompassing both natural and programming languages. Its universality as an ensemble approach heightens the performance of language models, agnostic of their underlying neural network architecture.

3. The empirical validation of this research is robust. The efficacy of the proposed method is rigorously assessed against multiple recognized benchmarks and contrasted with strong, widely accepted baselines.

**Weaknesses:**

1. One potential drawback of the proposed MIM model is its increased computational demand and memory usage, attributable to the incorporation of two models, as compared to conventional left-to-right LMs. It would enhance the comprehensiveness of the comparison if the MIM model is benchmarked against an ensemble of two left-to-right LMs pretrained with different random seeds. The experiment is useful even on a smaller scale.

2. The visual presentation of the paper could benefit from improvements. The majority of the tables and figures (except figure 1), are inadequately sized for legibility, particularly when printed. This could impede the reader's understanding and appreciation of the presented research.

**Questions:**

N/A

---

> ### Author Rebuttal · Authors · 2023-08-08
>
> We appreciate your thoughtful review and for recognizing the novelty and efficacy of our proposed method. We would like to address the weaknesses you identified and clarify our approach to addressing them.
>
> Weaknesses:
>
> Increased Computational Demand and Comparison with Ensemble of Two Left-to-Right LMs: We acknowledge the concern regarding the increased computational demand due to the bidirectional nature of our model. Your suggestion to compare MIM with an ensemble of two left-to-right LMs pretrained with different random seeds is insightful but may not offer a direct comparison. Such an ensemble would possess twice the number of parameters, unlike our MIM model, which utilizes one set of shared parameters. While we can explore including this experiment, we are unsure about its capacity to elucidate further the reasons behind MIM's superior performance. We will, however, commit to clearly detailing the computational requirements in the paper and provide a more nuanced discussion of how MIM compares with other approaches.
>
> Visual Presentation of Tables and Figures:  We value your feedback regarding the legibility of the tables and figures in our paper. We recognize that the visual presentation is crucial for a clear understanding of our findings. We will include comprehensive tables in full-size in the appendix and in the main paper we will provide more focused comparisons with only the most relevant baselines.

---

> > ### Comment · Reviewer_V8eN · 2023-08-11
> >
> > Thank your for your clarification. I would like to keep my original review score (7: Accept).

---

### Official Review · Reviewer_zFcY · 2023-07-06

**Soundness:** 3 good
**Presentation:** 3 good
**Contribution:** 3 good
**Rating:** 8
**Confidence:** 5

**Summary:**

The paper proposes a new pre-training paradigm for language models that jointly improves the training data efficiency and the capabilities of the LMs in the infilling task. The first technique is a training objective that aligns the predictions of a left-to-right LM with those of a right-to-left LM, trained on the same data but in reverse order. The second technique is a bidirectional inference procedure that enables both LMs to meet in the middle.

The paper reports extensive experiments on both programming and natural language models, outperforming strong baselines.

**Strengths:**

1. A nice idea to pretrain from both directions to improve data efficiency through consistency regularizer. The result models are naturally fit for infilling tasks.
2. Strong results compared to FIM baselines with 3+% gain on most tasks evaluated.
3. Clear ablations to understand where the gain comes from by turning off consistency regularizer.

**Weaknesses:**

1. Computational cost: training requires to train two models (forward and backward), which comes with additional computational cost.
2. Evaluation is restricted to coding. While there is perplexity eval on natural languages, it is unclear whether lower perplexity translates to gains in model capabilities such as few-shot learning.


**Questions:**

Quantify how much additional computational cost there is compared to regular left-to-right pretraining. Ablation of using such additional computation to train the left-to-right longer, and compare results to understand whether the gain reported in this paper comes from computational advantages.

---

> ### Author Rebuttal · Authors · 2023-08-08
>
> We are grateful for your comprehensive review and for recognizing the contributions and results of our paper. Your thoughtful comments have provided us with valuable insights, particularly concerning the computational costs and evaluation limitations. We'd like to address these points and clarify our intentions.
>
> Weaknesses:
>
> Computational Cost: Indeed training two models (forward and backward) incurs additional computational expenses. We acknowledge this and appreciate your interest in quantifying this aspect. We will include an analysis of the computational cost in comparison to regular left-to-right pretraining in the camera ready version of our paper. This will help clarify whether the gains we reported are attributed to computational advantages or inherent model efficiencies. Note that reviewer M9cS has raised a similar question so please check our response to them as well.
>
> Evaluation Limitation in Coding: While our focus was primarily on coding, your point about extending the evaluation to natural languages and examining downstream task performance is insightful. However such comparisons can be confounded by the exact in-context learning strategy and the amount of effort spent in prompt engineering. Our scope is pre-training and, similar to other works with the same scope, we mostly focus on perplexity and coding task performance.
>
> Questions:
>
> Ablation and Comparison with Extended Left-to-Right Training: Your suggestion to perform an ablation study using the additional computation to train the left-to-right model longer is particularly insightful. This will allow us to delineate whether the improvements stem from the bidirectional approach itself or merely from the extended training. We commit to including this analysis in the camera ready version as it will provide a more nuanced understanding of the strengths and potential limitations of our method.
>
> Your review has highlighted essential areas where we can further refine our work, and we appreciate your positive evaluation and constructive feedback. We are committed to incorporating these insights to enhance the clarity and robustness of our research.

---

> > ### Comment · Reviewer_zFcY · 2023-08-11
> >
> > Thank you for the clarifying comments and addressing my concerns.
> >
> > Regarding the ablations on training left-to-right model with 2x compute, I would like to see such ablations include in future revisions as it is such an important point to differentiate where the gain comes from.
> >
> > Regarding the few-shot evaluations and prompt engineering, I disagree with the statement that "such comparisons can be confounded by the exact in-context learning strategy and the amount of effort spent in prompt engineering". As long as we adopt the standard few-shot in-context learning for both baseline and the proposed method, it is robust enough to show the effect on model's actual capabilities. I want to emphasize again that lower perplexity doesn't always lead to better few-shot capabilities.
> >
> > I will keep my original review score, conditioned upon the completion of the suggested ablation and further evaluations, which I believe will make this paper more robust.

---

> > > ### Author Response · Authors · 2023-08-11
> > > **Our commitments**
> > >
> > > Upon further reflection, we see your point and agree that lower perplexity does not guarantee better few-shot capabilities.
> > >
> > > We commit to have these experiments in the camera-ready version:
> > > - We will compare MIM with training two left-to-right models
> > > - We will compare MIM models with other models on metrics besides perplexity and zero-shot evaluation.

---

### Official Review · Reviewer_SDkH · 2023-07-06

**Soundness:** 3 good
**Presentation:** 2 fair
**Contribution:** 2 fair
**Rating:** 5
**Confidence:** 4

**Summary:**

This paper proposes an extension of a paper  "Fill in the middle" (FIM)  (Bavarian et al 2022). The idea is to  define different objective function by considering two models for both directions and to "Meet in the middle" (MIM). One goal is to be more data efficient and the targeted application is in-filling for code generation.


**Strengths:**

This paper proposes an extension of a paper  "Fill in the middle" (FIM)  (Bavarian et al 2022). The idea is to  define different objective function by considering two models for both directions and to "Meet in the middle" (MIM). One goal is to be more data efficient and the targeted application is in-filling for code generation.


**Weaknesses:**

The method is simple, but the motivations remain unclear. For
instance, the paragraph on FIM should be improved. If a reader does
not already know about FIM, it is difficult to understand.  You could
provide some equations. Moreover a discussion, on why MIM is simpler
(and could be better) is necessary.  Moreover my main concern is about
the results: at the end why MIM is really more data-efficient that FIM
or than a BERT-like trained from scratch of the same task ?


**Questions:**

The first sentence of the abstract sounds really strange: the masked LM objective is used for instance with BERT, and the "B" means bidirectionnal. Beyond this lazy remark, a real discussion in the paper on this bidirectional point could be insightful for the reader. In the related work section the authors say that there is large body of work on bidirectional training. What are the real motivation of this new criterion ? What are the differences with BERT MLM, or ELMO ? What is expected in terms of performance ?

- The experimental part focuses on the comparison between FIM and MIM (in the text). The differences with other  models are not really commented ?
- The models used could be better described: "a decoder only transformer LM". Could you precise the number of layers, heads, ... (maybe in an appendix) ?


**Limitations:**

Not really.

---

> ### Author Rebuttal · Authors · 2023-08-08
>
> Thank you for taking the time to thoroughly review our paper. Your comments have highlighted some important areas for clarification and improvement, and we are grateful for the opportunity to address your concerns.
>
> Weaknesses:
>
> Unclear Motivation: The motivation for our work stems from addressing the limitation of FIM and pure left-to-right training. We discussed some of these in the first three sections of the paper but space limitations constrained a more comprehensive discussion. We refer readers to the papers we cite in the related work section in particular the Fill in the Middle paper and the Twin Networks paper, for additional insights.
>
> Comparison with FIM and BERT on Data Efficiency: Your concern about why MIM is more data-efficient than FIM or a BERT-like model trained from scratch for the same task is well-noted. Many tasks we are targeting are challenging for BERT due to its non-autoregressive nature. Concerning FIM, MIM's advantage stems from keeping the context close to the generation, which we believe leads to its outperformance over FIM. We also explained this in our response to reviewer 5x9n.
>
> Questions:
>
> Discussion on Bidirectional and Comparison with BERT and ELMO: We appreciate your suggestion to discuss the bidirectional aspect more thoroughly.
> The real motivation behind our work is rooted in the explosive adoption of generative language models for a wide array of use cases, including chatbots and assistive co-authoring. Unlike BERT and ELMO, where generation is non-trivial, FIM and MIM excel in text or code infilling. Our work shows that for such generative tasks, bidirectional modeling during training can still help.
>
> Comparison with Other Models besides FIM: We concentrated on FIM as it was our strongest baseline and it was trained with exactly the same data as MIM. This helps isolate the effect of our proposal, while a comparison with other models would be confounded by other choices. We will include some more recent models in the camera ready version but the apples-to-apples comparison will remain FIM vs. MIM.
>
> Model Details: We appreciate your call for a more precise description of the models used, including the number of layers, heads, etc. We agree that this information is valuable, and we will include it in the appendix.
>
> Thank you again for your thoughtful comments. We hope your concerns have been addressed by this response. If so, please consider raising your rating.

---

> > ### Comment · Reviewer_SDkH · 2023-08-11
> >
> > Thank you for the clarification. The overall score was maybe too low, but I still have concerns with this work. I think it's a nice idea, but maybe the contributions are a bit below what we could expect at NEURIPS.
> >
> > Moreover, the computational cost must be discussed as well.

---

> > > ### Author Response · Authors · 2023-08-11
> > > **Clarifications**
> > >
> > > Thank you for raising your score.
> > >
> > > We would like to clarify that based on our responses to the other reviewers, we have committed to analyze and discuss the computational cost in detail in the camera-ready version.
> > >
> > > Regarding the strength of the contributions, one thing we neglected to mention is that we are going to release MIM model checkpoints and inference code upon acceptance. Moreover, the technique we propose is composable with many other pre-training techniques (e.g. dataset preparation) which we think can lead to wide adoption. As pre-training is so foundational for LLMs, the effects of consistent improvements at this stage can benefit many downstream tasks. To sum up, we believe the impact of this work can be significant.

---

### Official Review · Reviewer_5x9n · 2023-07-06

**Soundness:** 3 good
**Presentation:** 3 good
**Contribution:** 3 good
**Rating:** 7
**Confidence:** 4

**Summary:**

The paper proposes a new pretraining objective called meet in the middle. This is a bidirectional objective consisting of forward plus backward logprob and an agreement term measured by total variation distance between the two. The forward and backward models share parameters and optionally attention.

Inference is done by generating in two directions until they meet in the middle with agreement, potentially introducting a 2x factor in parallelism and reduced latency.

The model are tested on both code generation accuracy and perplexity on several training set and the pile. They compared to both published baselines training on different data as well as FIM trained on the same data, showing strong results.

**Strengths:**

bidirectional agreement objective that is pretty simple and intuitive in some ways, giving strong results compared to FIM. should be easy enough for others to adopt

results and comparisons are strong

some helpful ablations are included

**Weaknesses:**

"New paradigm" is perhaps exaggerated for introducing bidirectional model via sharing and agreement

still more complexity compared to FIM, and some source of improvements are not very clear. For instance, there are several claims of improved data efficiency due to bidirectionally, but the experiments do not seem to test this directly. If this is the main claim, then some direct left to right comparisons would make more sense.

the optional improvements seems pretty important and yielded half the improvement over FIM, so perhaps should not be considered optional. For instance, HE infilling results for MIM, MIM-lambda=0, and FIM are respectively 26.3, 24.7 and  22.8. The caption of table 1 says the 26.3 was without the enhancement from 3.2.2 whereas the ablations in table 5 suggest it was.

no comparisons with more recent models like santacoder, starcoder



**Questions:**

given that FIM actually conditions on both the prefix and suffix whereas MIM only conditions on one at a time with weak interactions, any discussion on how MIM outperforms despite that?

not very clear what is the backup when the model fails to meet in the middle and how false positives are dealt with, especially for long context generations.

**Limitations:**

Yes

---

> ### Author Rebuttal · Authors · 2023-08-08
>
> Thank you for your thorough review and insightful comments on our paper. Your observations have guided us to reflect on certain aspects of our work, and we appreciate the opportunity to clarify and expand on those points.
>
> Weaknesses:
>
> Testing Improved Data Efficiency Directly: Your observation concerning the lack of direct evidence for improved data efficiency is well-noted.  We believe that the results in Table 4 provide indirect support for our claim, particularly considering the empirical support of the claim that "FIM is for free/cheap" in the original FIM paper (Bavarian  et. al. 2022) as well as in other papers including Santacoder and Starcoder that you mentioned. However, we acknowledge that a direct comparison with a non-FIM model might have been more illustrative. We will include this comparison in the camera ready version.
>
> Wording on Optional Improvements: Indeed these improvements are only optional for left-to-right generation. We will change the wording to "infilling improvements" to be more precise. We are grateful to you for highlighting the error in Table 1's caption. We will ensure to correct it. To clarify, the columns in Table 1 that summarize the infilling tasks use the model with infilling improvements.
>
> Comparisons with Santacoder and Starcoder: We appreciate your interest in comparisons with these recent models. Both of these models were trained with the FIM approach (Bavarian et. al. 2022). MIM is a general technique whose purpose is to improve upon FIM. Therefore the proper comparison is whether Santacoder/Starcoder trained with MIM outperforms Santacoder/Starcoder trained with FIM. We will explain this point in more detail in the camera ready version.
>
> Questions:
>
> How MIM Outperforms FIM: We speculate that MIM beats FIM because of context placement. In FIM, either the prefix or the suffix must be placed further from the completions, while in MIM, the relevant context is always nearby. This placement likely provides MIM with an advantage in attending to relevant information and achieving better performance.
>
> Backup When the Model Fails to Meet in the Middle: In such cases, we resort to using the completion with the best probability according to the model that generated it. This is explained in line 140 of the paper.

---

> > ### Comment · Reviewer_5x9n · 2023-08-11
> >
> > Thanks for the response and discussions. Will keep the score.

---

### Official Review · Reviewer_M9cS · 2023-07-07

**Soundness:** 3 good
**Presentation:** 4 excellent
**Contribution:** 3 good
**Rating:** 6
**Confidence:** 5

**Summary:**

This paper presents a new pre-training approach called MIM for language models, which promotes agreement between left-to-right and right-to-left models regarding their token distribution for each position. The authors have developed a straightforward and effective infilling inference method that utilizes context from both directions and the tendency of forward and backward models to align. The paper pre-trains language models of different sizes on public code and language data using MIM, and the results of various experiments demonstrate the superiority of MIM over existing baselines.

**Strengths:**

- This paper investigates the bidirectional agreement approach for language models, and the results of various experiments demonstrate the superiority of MIM over existing baselines.
- The methodology employed in the paper is robust, and the results are statistically significant. The ablation studies conducted in the paper confirm the effectiveness of the primary proposals during both training and inference.

**Weaknesses:**

- The proposed method is not very novel, as similar ideas have been proposed in the NMT field (treated as conditional language models). However, it is still valuable to investigate the impact of these methods on pure language models.
- The bidirectional agreement approach leads to more memory cost and training time.

**Questions:**

1. Have you experimented with MIM using larger model scales, such as 7b and 13b? Have you identified any scaling laws for this approach?
2. The predicted probability distribution from different directions may not the identical due to the different contexts. For instance, predicting the first token from a left-to-right perspective may be challenging, but it may be more accurate when using a right-to-left approach. What is your perspective on this? Could you provide more information on the probability differences observed in various positions?
3. Some of the methods previously used in the NMT field may be directly used for language models. Have you evaluated the performance of these methods in your experiments?

---

> ### Author Rebuttal · Authors · 2023-08-08
>
> Thank you for taking the time to provide a comprehensive review of our work. We appreciate your thoughtful feedback and constructive comments. Below, we address the concerns and questions you raised.
>
> Weaknesses:
>
> More Memory Cost and Training Time: Our method indeed requires more memory and training time. However, the memory consumption can be effectively mitigated by using a smaller batch size during training and we observed that the smaller batch size does not harm the training convergence. Furthermore, since our two models share parameters, the overall memory footprint is smaller than having two separate models. We believe that the benefits and robustness of the proposed method make these additional computational costs justifiable and we will include a discussion of these costs in the camera ready version.
>
> Questions:
>
> Larger Model Scales: Unfortunately, due to limited resources, we have not pre-trained 7B and 13B parameter models. We will try to perform scaling law experiments on small scale models which can suggest how perplexity will extrapolate to these larger models by the camera ready deadline.
>
> Probability Distribution Differences: You astutely noted the potential for differences in the predicted probability distribution due to varying contexts. We concur that in many cases,
> $p(x_i | x_{<i})$ should not be equal to $p(x_i | x_{>i})$. Our regularizer's design indeed does not place a special emphasis on the actual token $x_i$ but aims to minimize the average difference across all words in the vocabulary and across all positions. The regularizer does not need to be universally correct, it just needs to have a positive contribution which is demonstrated by our empirical results.
>
> Methods from the NMT Field: We agree that integrating ideas from neural machine translation could lead to intriguing advancements in pre-training. Our work takes inspiration from some bidirectional approaches in the NMT field that have led to substantial improvements. However, exploring further techniques in this paper is beyond our scope.

---

> > ### Comment · Reviewer_M9cS · 2023-08-11
> >
> > Thanks for your clarification. I would like to keep my score.

---

### Decision · Program_Chairs · 2023-09-21

**Decision:**

Accept (poster)

**Comment:**

The paper introduces "Meet in the Middle" (MIM), a new pre-training paradigm for autoregressive language models that trains in both left-to-right and right-to-left directions to improve data efficiency and performance in tasks like infilling.

Reviewers generally agree that the MIM approach is robust, with positive results outperforming existing baselines, specifically Fill in the Middle. The methodology is considered simple, intuitive, and easily adoptable (M9cS, 5x9n). MIM's applicability to both natural and programming languages is praised, as are the clear ablation studies that help understand its effectiveness (SDkH, zFcY, V8eN).

Reviewers, however, raised concerns about the increased computational cost and memory usage due to the bidirectional aspect (M9cS, zFcY, V8eN). Some reviewers question the novelty, pointing out that similar ideas exist in Neural Machine Translation (M9cS, 5x9n). There's also a call for clearer motivation and justification, especially about MIM's data efficiency compared to existing methods like FIM and BERT (SDkH, 5x9n).